# Effect of Increased Cabin Recirculation Airflow Fraction on Relative Humidity, CO$_2$ and TVOC

Victor Norrefeldt [1,*], Florian Mayer [1], Britta Herbig [2], Ria Ströhlein [2], Pawel Wargocki [3] and Fang Lei [3]

1 Fraunhofer Institute for Building Physics IBP, 83626 Valley, Germany; florian.mayer@ibp.fraunhofer.de
2 Institute and Clinic for Occupational, Social and Environmental Medicine, LMU University Hospital Munich, 80336 Munich, Germany; Britta.Herbig@med.uni-muenchen.de (B.H.); Ria.Stroehlein@med.uni-muenchen.de (R.S.)
3 Department of Civil Engineering, Technical University of Denmark, 2800 Kgs. Lyngby, Denmark; paw@byg.dtu.dk (P.W.); fl@byg.dtu.dk (F.L.)
* Correspondence: victor.norrefeldt@ibp.fraunhofer.de

**Abstract:** In the CleanSky 2 ComAir study, subject tests were conducted in the Fraunhofer Flight Test Facility cabin mock-up. This mock-up consists of the front section of a former in-service A310 hosting up to 80 passengers. In 12 sessions the outdoor/recirculation airflow ratio was altered from today's typically applied fractions to up to 88% recirculation fraction. This leads to increased relative humidity, carbon dioxide (CO$_2$) and Total Volatile Organic Compounds (TVOC) levels in the cabin air, as the emissions by passengers become less diluted by outdoor, dry air. This paper describes the measured increase of relative humidity, CO$_2$ and TVOC level in the cabin air for the different test conditions.

**Keywords:** aircraft air quality; adaptive ECS; subject testing



## 1. Introduction

The environmental control system (ECS) is one of the major secondary energy consumers of an aircraft, consuming 3–5% of the power produced by the engines [1]. As outside conditions in cruise altitude are life threatening, the environmental control system fulfils a lifesaving function for the passengers by maintaining cabin pressurization and furthermore maintains a comfortable temperature and air quality.

In most commercial aircraft, the outdoor air used to ventilate the cabin is extracted from engine compressor stages upstream of the combustion chamber. Due to the compression, this air, called bleed air, is hot and its temperature exceeds 200 °C. An expansion to cabin pressure and cooling by ram air bring down temperature to an adequate level. In the mixing chamber, this outdoor air is mixed with HEPA (High Efficiency Particulate Arresting) filtered recirculation air. From there, the air supply passes through the riser ducts to the cabin air outlets, from where it enters the cabin [2].

Despite the low outdoor temperatures in cruise (below −50 °C) and the relatively thin layer of insulation between cabin and skin (order of 2–4 inches), the high passenger density usually leads to a cooling requirement for the cabin. Thus, the ECS primarily supplies air below cabin temperature. The moisture in the cabin is typically low, because the outside air is cold and thus dry. The major source for humidification are the passengers themselves, emitting water by breathing and sweating. Along with this, passengers exhale CO$_2$ and emit VOCs (volatile organic compounds) as a product of metabolism. Toiletry and cleaning agents are additional sources of VOCs in the cabin. These compounds are diluted by the outdoor air, while the recirculation airflow does not alter the composition. Aircraft ECS are required to comply with target values for both pressure and temperature. Typical values for the requirements are listed below [3].

- Minimum cabin air pressure limited to 750 hPa (equivalent to pressure at an altitude of 8.000 ft. or 2.400 m above sea level)
- Temperature range in the cabin between 18.3 and 23.9 °C (65 to 75 °F)
- Minimum outside airflow rate per passenger of 3.5 L/s (7.5 cfm)
- Recommendation of 9.4 L/s (20 cfm) total flow, to be met by outside and filtered recirculated air

The bleed air offtake reduces the efficiency of the engine because some compressed air is extracted rather than contributing to fuel burn and thrust production. Therefore, a reduction of bleed air is associated with lower fuel burn. In the building sector, demand controlled ventilation already today implements a similar philosophy based on reducing the supply airflow rate to that actually required, thereby avoiding excessive ventilation and thus energy use. Often, $CO_2$ concentration in the occupied space is used as a feedback signal for the control of the ventilation system as it is closely correlated with other human emissions and the occupation density [4]. DIN EN 15251 [5] gives quality criteria for the indoor air based on $CO_2$ level and states that concentrations below 800 ppm are optimal and concentrations above 1.200 ppm are not acceptable. Whether these design values can be directly applied to aircraft is not clear because buildings typically do not have HEPA filtration in the recirculation air, whereas this is standard for aircraft. A comparison of building and aircraft ventilation requirements is provided in [6].

A series of measurements taken during commercial flights by Giaconia et al. [7], found that average $CO_2$ concentrations varied between 925 and 1449 ppm. Unfortunately, the publication is not clear on whether the $CO_2$ readings were compensated for pressure or not. Typically, a pressure correction needs to include the factor $1.013/p_{cabin}$. [8]. In a similar approach on 179 domestic US flights Cao et al. [9] found an average $CO_2$ concentration of $1353 \pm 290$ ppm in the cabin. The researchers clearly state that sensor readings were pressure compensated. At the current stage, a review paper on cabin air quality is being prepared by the authors. This review confirms the magnitude of $CO_2$-concentrations reported.

In this context, the ComAir study was proposed to further investigate aircraft ventilation strategies and the impact of indoor air quality on passengers' comfort and well-being.

In today's aircraft, the adjustment of the ECS to the passenger count is limited to discrete, manual steps, like switching off an ECS pack or running it on low or high setting, however no controlled adaption of the flow is implemented. The research performed in the ComAir project shall give the basic design parameters for the possible future implementation of an adaptive ECS. This system is investigated in a CleanSky2 project [10]. For this, the impact of decreased outdoor air intake and increased recirculation flow rate is investigated in a subject study. Half full and fully booked conditions are investigated to assess the effect of passenger count.

The ComAir study was approved by the Ethics committee of the Faculty of Medicine, Ludwig-Maximilians-University, Munich (ID: 19-256) and written informed consent was obtained from all participants. The tests were performed in November 2019 and thus before the Sars-Cov-2 pandemic was an issue in Germany. This paper summarizes the resulting $CO_2$, humidity and TVOC levels measured in a cabin mock-up for the different outdoor air intakes.

## 2. Method

To investigate the effect of reduced outdoor air intake and increased recirculation airflow rate, a randomized controlled study with a total of 559 different participants was conducted. The study was hosted in the Flight Test Facility (FTF, https://www.hoki.ibp.fraunhofer.de/vr/virtual-tour_IBP/) located at the Fraunhofer-Institute for Building Physics IBP in Holzkirchen, Germany.

### 2.1. Flight Test Facility Test Setup

In the FTF, a former in-service front section of an A310 is located in a low pressure vessel (Figure 1a). The cabin (Figure 1b) hosts up to 80 passengers and is surrounded by crown, forward galley, cockpit, avionics, triangles, cargo and bilge compartments. The aircraft mock-up consists of the section from nose to front wing box wall (Figure 1c). The ECS is emulated by a building type air conditioning system with a sub freezer that cools and dehumidifies the supply air to a dew point of −20 °C. Thus, a similar low humidity level of supply air can be reached as in flight. Air was supplied through ceiling outlets above the stowage bin and lateral outlets that have been added to better emulate today's cabin design. The recirculation air is extracted from the cabin and HEPA filtered with original filters. A schematic view of the airflow path is shown in Figure 1d.

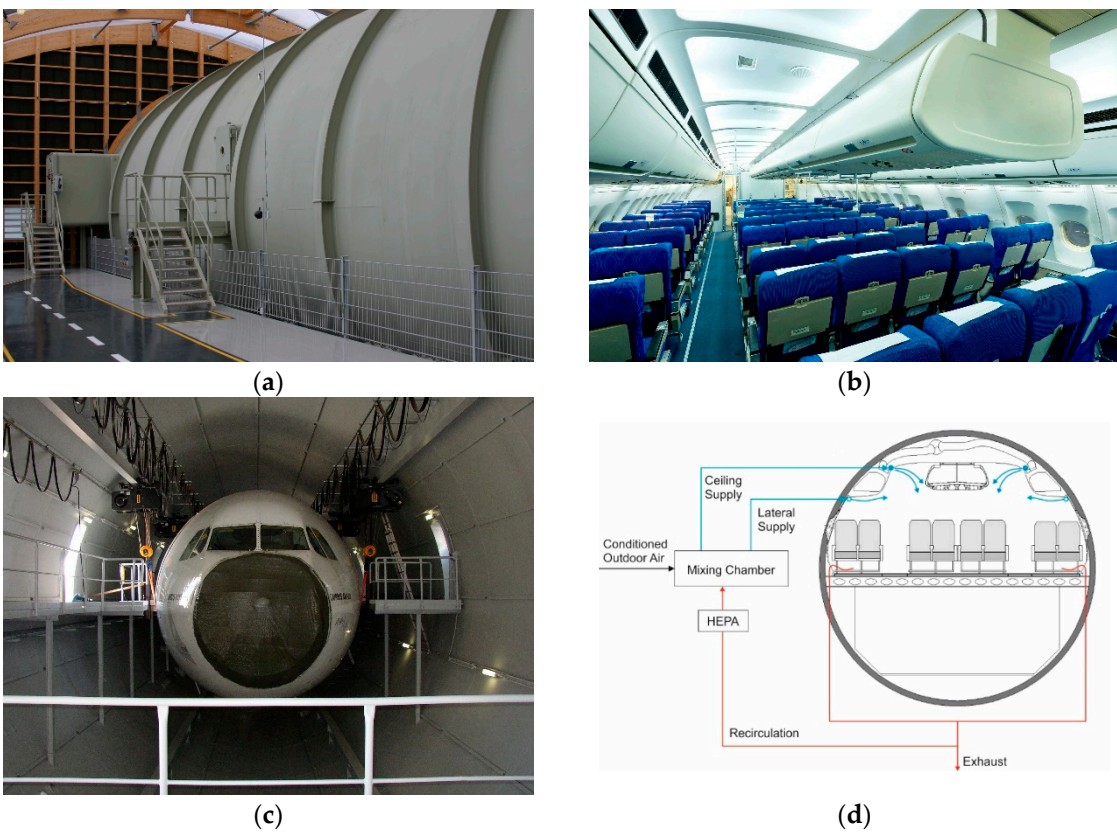

**Figure 1.** (**a**) Low pressure vessel of FTF, (**b**) Aircraft cabin mock-up, (**c**) Aircraft mock-up in the low-pressure vessel, (**d**) schematic ventilation pattern.

In the cabin, the temperature stratification was measured at eight locations at heights of 0.1, 0.6, 1.1, 1.7 and 2.2 m. $CO_2$ and relative humidity were measured at a height of 1.1 m at these locations (Figure 2). VOCs are pumped from the front left measurement location on sample tubes and later analyzed by GC-MS (gas chromatography-mass spectrometry) or by HPLC-DAD (high performance liquid chromatography with diode array detector) in the laboratory. $CO_2$ sensors were calibrated with 4.000 ppm calibration gas both at ground pressure (~940 hPa for Holzkirchen due to the place's elevation) and low pressure of 755 hPa in the vessel and a pressure correction was derived. The outdoor and recirculation airflow rates were measured with anemometers; a sufficient length of the straight pipe section to ensure accurate measurement was ensured. The following sensor types and specifications were used:

- Temperature: Four-wire PT100 thermocouples with an accuracy ±0.1 K @ 20 °C according to DIN EN 60751 [11] class A.
- Humidity: Rotronic HygroClip HC2-C05 sensor with ±1.5% RH [12]

- $CO_2$: Vaisala GMW20, range: 0–5.000 ppm, accuracy ±2%. Sensors were calibrated with 4.000 ppm calibration gas both at ground pressure (~940 hPa for Holzkirchen due to the place's elevation) and low pressure of 755 hPa in the vessel and a pressure correction was derived.
- Pumped tubes: samples drawn for 20 to 60 min with flow rates of 0.1 to 1.0 l/min (depending on target compounds) and analyzed by GC-MS (gas chromatography-mass spectrometry QP2010 SE, Shimadzu, Duisburg, Germany) or HPLC-DAD (high performance liquid chromatography with diode array detector, Agilent 1260 Infinity, Agilent Technologies, Waldbronn, Germany). VOCs were analyzed according to DIN ISO 16000-6 [13] and carbonyl compounds according to DIN ISO 16000-3 [14].
- Flow rate: Schmidt SS20.500 Sensors 0–35 m/s with an accuracy of ±3% [15].

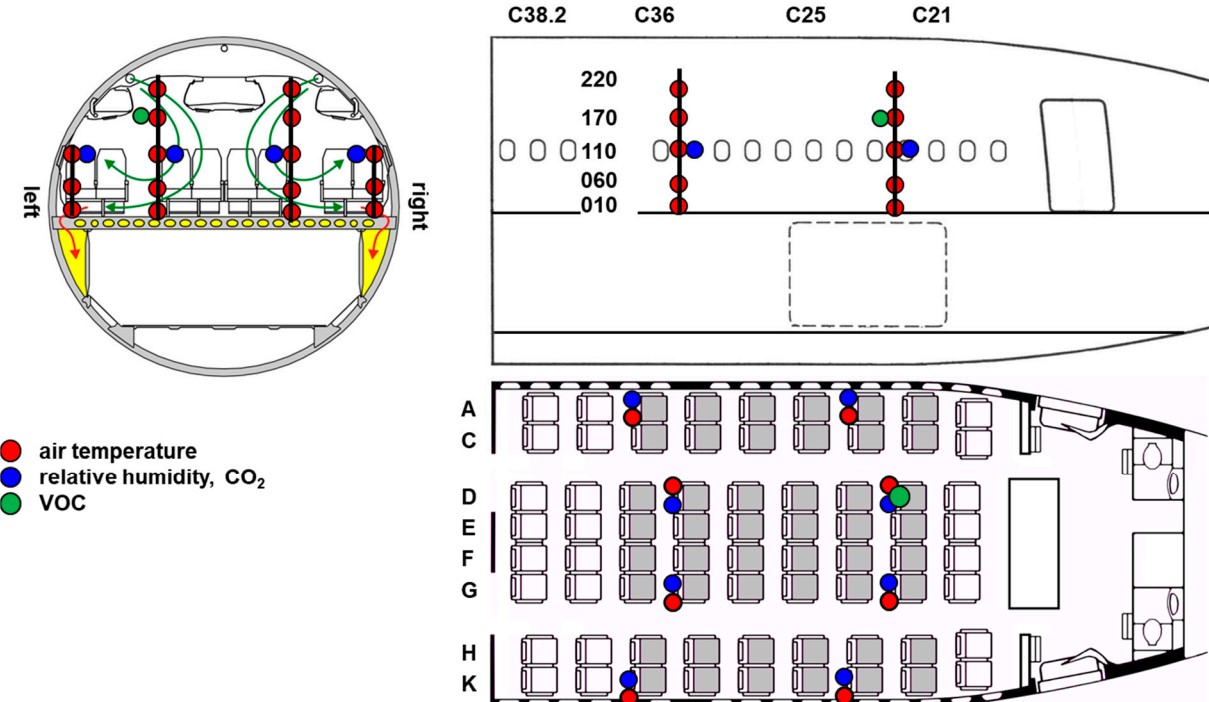

**Figure 2.** Cabin measurement locations.

## 2.2. Test Matrix and Sequence

The study investigated four outdoor airflow rates and two occupancy levels: uncongested/half and fully booked cabin (Table 1). In order to expose a similar number of subjects to each condition, the uncongested conditions were tested twice while the fully booked conditions were each tested once. Subjects were blinded to experimental condition and were only allowed to participate in one test, thus, nobody witnessed two conditions. The lower outdoor airflow rates were compensated with increased recirculation airflow rates in order to maintain a constant total flow rate per passenger of 9.4 L/s. The flow rates were adjusted for each test based on the actual count of subjects in the cabin. As a result, the fully booked conditions had a higher flow rate at the cabin air outlets than the uncongested conditions. Cabin temperature set-point was 23 °C in order to maintain a thermally comfortable situation. When setting up the test matrix, the following considerations were made:

- Baseline: Replication of today's typical $CO_2$ levels reported in aircraft
- ASHRAE: Replication of the minimum requirement for outdoor airflow rate (3.5 L/s/passenger) set out by ASHRAE 161

- ASHRAE half: Half the required outdoor airflow rate (1.8 L/s/passenger). Because the $CO_2$ level follows the inverse of the outdoor airflow rate, this point was chosen because it was pre-assessed to be in the middle between the ASHRAE and the Max. $CO_2$ condition.
- Max. $CO_2$: Lowest outdoor airflow rate designed to remain below 5.000 ppm limit [16].

**Table 1.** Test Matrix.

| Conditions | Baseline | ASHRAE | ASHRAE Half | Max. $CO_2$ |
|---|---|---|---|---|
| Outdoor airflow rate in L/s/passenger | 5.2 | 3.5 | 1.8 | 1.1 |
| Recirculation airflow rate in L/s/passenger | 4.2 | 5.9 | 7.6 | 8.3 |
| Total airflow rate in L/s/passenger | 9.4 | 9.4 | 9.4 | 9.4 |
| Fully booked (~70–80 PAX) | 1 session | 1 session | 1 session | 1 session |
| Uncongested (~35–40 PAX) | 2 sessions | 2 sessions | 2 sessions | 2 sessions |

The following names will be used for the different test cases:

- Baseline–uncongested 1 (1st Session)
- Baseline–uncongested 2 (2nd Session)
- Baseline–fully booked
- ASHRAE–uncongested 1 (1st Session)
- ASHRAE–uncongested 2 (2nd Session)
- ASHRAE–fully booked
- ASHRAE half–uncongested 1 (1st Session)
- ASHRAE half–uncongested 2 (2nd Session)
- ASHRAE half–fully booked
- Max. $CO_2$–uncongested 1 (1st Session)
- Max. $CO_2$–uncongested 2 (2nd Session)
- Max. $CO_2$–fully booked

The test sequence is shown in Figure 3. Subject reception and medical pre-screening was 1h before the test started. After boarding, the pressure in the chamber was reduced (Figure 4). When flight altitude was reached, the desired ventilation regime was set. Subjects answered psychological and health questionnaires and were tested for performance in Batteries 1 and 2, and detailed questionnaires on comfort were distributed in the middle and at the end of the session (Comfort Battery and Battery 2). After test battery 2 the cabin was repressurized and subjects deboarded.

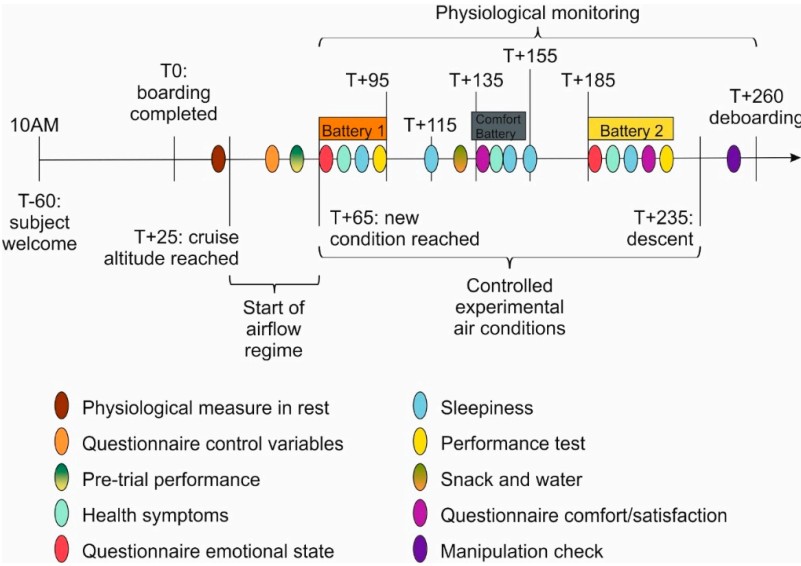

**Figure 3.** Test sequence.

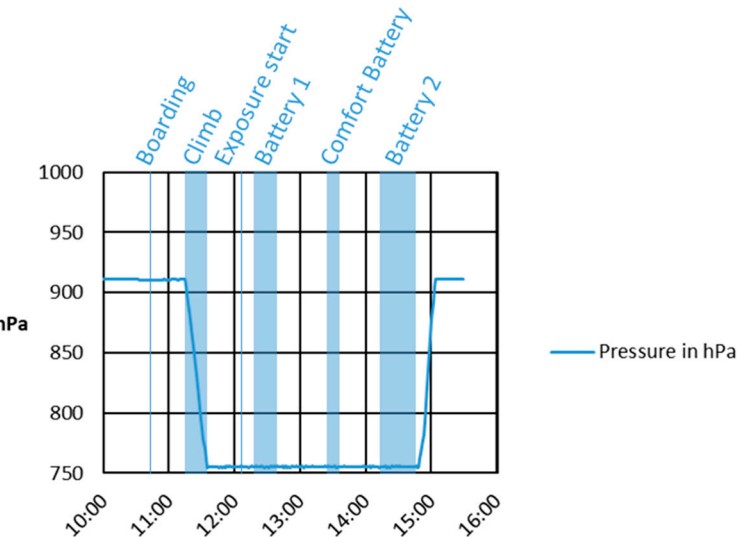

**Figure 4.** Example of Cabin pressure profile.

Only low emitting food (pretzels) and water were served to the subjects to not artificially alter the VOC composition in the cabin. In a real flight, coffee, tea, juices, alcoholic drinks and food with higher emission may be served as well. While for the high outdoor airflow rates, these events are expected to be flushed out quickly from the cabin, the low outdoor airflow rates show noticeable transient times and thus the effect of such events would persist longer. Furthermore, possible exhaust ingestion or engine oil (fume) events were disregarded in this study as it focused on emissions generated inside the cabin.

### 2.3. Assessment of Air Quality

The assessment of the indoor environment was performed by a trained sensory panel according to the requirements of ISO 16000-30 [17] and subjects' ratings via questionnaire at the end of the session.

For the first test of each condition, a sensory panel consisting of eight trained test persons entered the cabin and assessed the perceived intensity, the hedonic odor tone and the odor quality of the cabin air. Their evaluation of perceived intensity (pi) uses of a comparative scale, that assigns the odor intensity of air to a defined concentration of acetone (Figure 5). Hedonic odor tone is assessed on a 9 point category scale from −4 (very unpleasant), via 0 (neither pleasant nor unpleasant) to +4 (very pleasant).

Within the set of questionnaires at the end of exposure (i.e., before descent), all subjects were asked to:

- rate the smell in the cabin on a five point scale (How would you assess the odor intensity in this flight? no odor; slight odor; moderate odor; strong odor; overwhelming odor),
- evaluate the air quality with a five point Likert scale (How would you rate the air quality in this flight? very poor; poor; average; good; very good/excellent)
- assess the acceptability of the air quality with an approach used by [18,19].

Overall, 559 persons representative of flight passengers with regard to sex and age (283 men, 276 women; mean age 42.68 ± 15.85 years) participated in the study and rated these parameters.

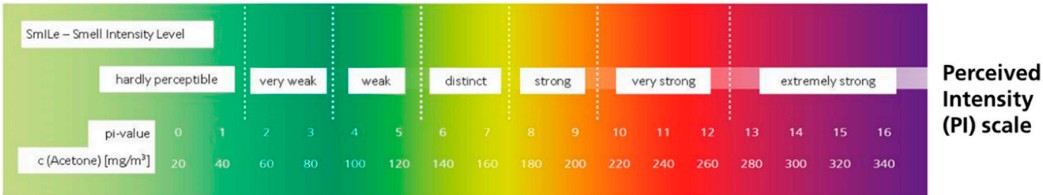

**Figure 5. Top**: Perceived intensity scale, **Bottom**: Hedonic odor tone scale.

*2.4. Test Preparations*

2.4.1. Considerations on Heat Balance

Based on the test matrix, some preliminary rule of thumb considerations were made on the heat balance of the cabin. As cabin cooling is provided by conditioned outdoor air, it was investigated to which extent the reduced flow of outdoor air is capable to fulfil this function. In this balance, the recirculation air is not considered, as from a thermal point of view it is extracted and re-injected into the cabin. For reasons of simplicity, it was assumed that each passenger brings in 100 W of heat by metabolism, electrical devices and in-flight entertainment. Neglecting heat losses through the envelope and heat gain from e.g., fan operation, the required outdoor air temperature entering the mixing box can be estimated by the equation below. Results of this assessment are summarized in Table 2. Especially for outdoor flow rates below the ASHRAE requirement of 3.5 L/s, the required temperature to maintain the cabin thermal balance becomes unrealistically low. Reaching such low temperatures would require a large ram air heat exchanger and would only be possible during cruise phase. Therefore, it was concluded that a recirculation heat exchanger connected to a mobile cooling machine will be needed for the experimental campaign. Based on the occupancy of 80 passengers, it was estimated that this cooling unit needs to extract up to 8 kW of heat assuming a close to zero cooling capacity of the lowest outdoor airflow rate of 1.1 L/s.

$$T_{supply} = T_{cabin} - \frac{\dot{Q}_{pax}}{\dot{V}_{outdoor} \cdot \rho \cdot c_p}$$

with: $T_{supply}$: Supply air temperature of outdoor air in °C, $T_{cabin}$: desired cabin air temperature (23 °C), $Q_{pax}$: Heat emission per passenger (100 W), $V_{outdoor}$: Outdoor airflow rate per passenger in L/s (norm conditions), $\rho$: air density (1.2 kg/m$^3$ at norm conditions) and $c_p$: 1004.5 J/(kg·K).

**Table 2.** Estimated requirement for outdoor air temperature and estimated cooling power required in the recirculation air path.

| Conditions | Baseline | ASHRAE | ASHRAE Half | Max. CO$_2$ |
|---|---|---|---|---|
| Outdoor airflow rate in L/s/PAX | 5.2 | 3.5 | 1.8 | 1.1 |
| Required outdoor air temperature | 7 °C | −1 °C | −23 °C | −52 °C |
| Recirculation cooling power | n/a | n/a | 4 kW | 8 kW |

2.4.2. Subject Safety

High priority was given to the safety of subjects in the cabin. Therefore, a detailed safety and hazard assessment was performed prior to the subject tests.

To ensure a safe conduction of the tests, a review of fire escape procedures was performed. Unnecessary equipment resulting in burn load was removed and smoke protection masks were purchased in sufficient number. It was ensured that fire extinguishers were in place and had a valid seal of approval. Escape maps were reviewed and updated. Prior to the test conduct, subjects were informed in the use of the smoke masks and the location of fire escape lanes, similar to the floating vest and exit instructions on a commercial flight.

It was ensured that at least one of the cabin crew had passed a recent first aid course. Furthermore, a medical doctor was onsite during the entire test carrying a medical emergency valet with him/her.

Prior to the test campaign, the test setup as well as the possibility to enter and leave the cabin through a pressure lock during low pressure operation was inspected by the local emergency rescue team.

The expected exposure in terms of $CO_2$ and VOCs was pre-assessed and it was assured that no regulatory limits were exceeded like e.g., FAR part 25, ref [16].

## 3. Results

### 3.1. Flow Rates

Figure 6 shows the flow rate per passenger. Obviously, the outdoor flow rate was maintained both for the uncongested and the fully booked cases. The total flow rate was close to 9.4 L/s/passenger for the tests. In the cases "ASHRAE uncongested 2" and "ASHRAE half uncongested 2" a slightly higher recirculation flow rate was applied because only a lower number of subjects participated (35 and 34 instead of 40). For the fully booked "Max. $CO_2$" case, the recirculation fan came to its limit resulting in slightly lower flow rate.

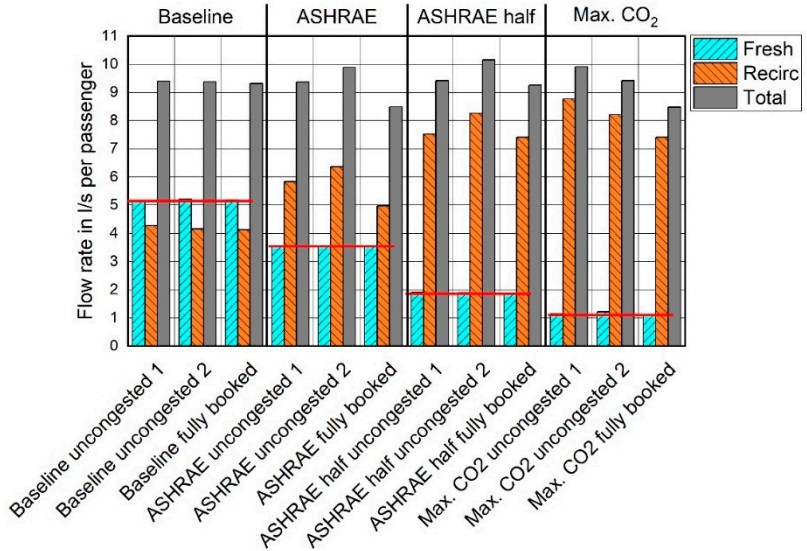

**Figure 6.** Passenger based airflow rates in the cabin.

Figure 7 shows the total airflow rates in the cabin and thereby proves the adaptive nature of the selected study approach. Due to the day by day adjustment of the airflow rate to the passenger count, each day results in different total flow rates and the fully booked cases have a noticeably higher total flow rate.

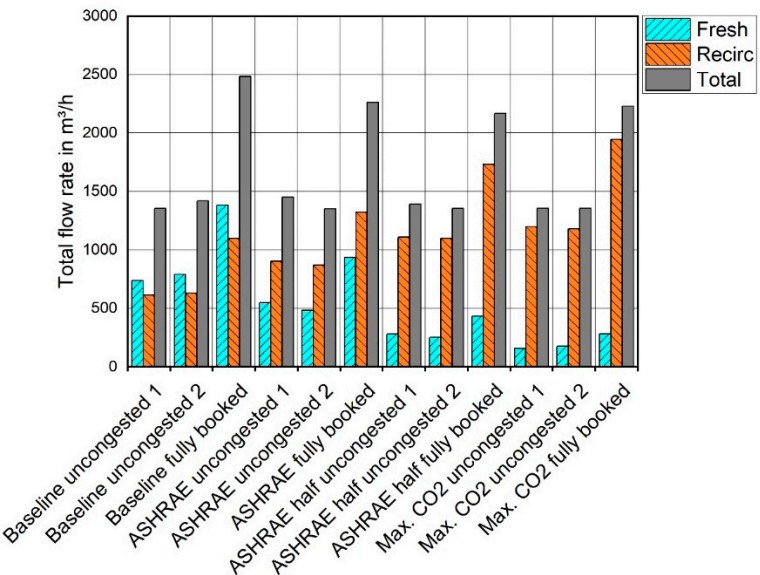

**Figure 7.** Total airflow rates in the cabin.

### 3.2. Temperatures

Temperatures were averaged for the entire exposure and for the timeslots the subjects filled out questionnaires (Figure 8). The cabin temperature and stratification usually were around ±1.5 K and thus below the limit of 2 K for a class A rating according to DIN EN ISO 7730 [20], therefore only the average temperature in the occupied zone (0–1.1 m height) is reported here. The temperature target of 23 °C was mostly maintained at ±1 K. It can be inferred by the measurements that the "Baseline fully booked" condition results in a colder cabin temperature because subjects were close to the control feedback sensor in the cabin and sufficient cooling power to react was available. For the "Max. $CO_2$ fully booked" condition, the available cooling power of the recirculation heat exchanger came to its limit and therefore temperature in the cabin was higher. For this case, a larger cooling system than the available one would have been necessary, possibly due to the additional heat injected by the recirculation fan operating at full speed.

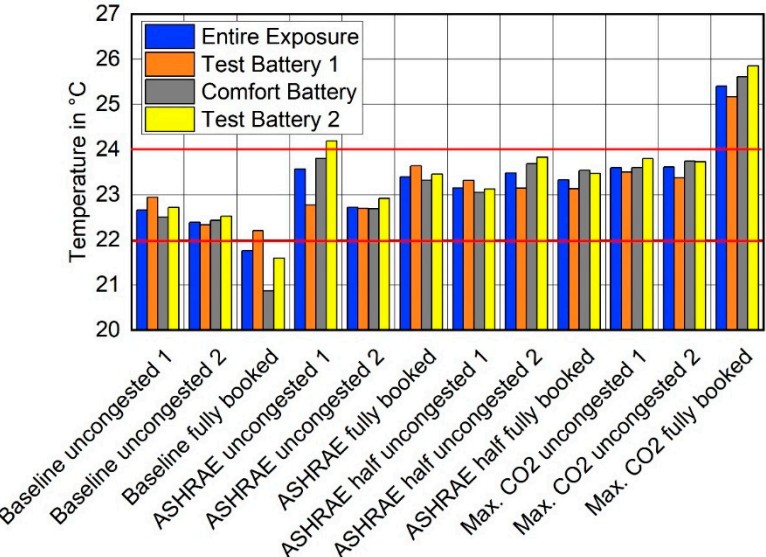

**Figure 8.** Average cabin temperature.

### 3.3. Cabin Humidity

A clear trend of increased cabin relative humidity with decreasing outside airflow rate is obvious (Figure 9). In the "Baseline" conditions, values around 15% RH were measured that are well in line with other publications [7,9]. For the "ASHRAE half" and "Max. $CO_2$" conditions, a gradual increase of relative humidity from test battery 1 over comfort battery to test battery 2 could be noticed. Due to the low outside airflow rate, the time constant of the cabin to reach steady-state was higher and thus the build-up phase time increased. Even at the end of the test, an increase in moisture was still obvious. For the fully booked case, the total flow rate was higher, and therefore humidity level was closer to convergence for the "Max. $CO_2$ fully booked" case, compared to the uncongested case at which a build-up could still be seen at the end of the test (Figure 10). Considering test battery 2 to be representative of the steady-state conditions reached at the end of the tests for all the experiments, a maximum value of 33.7% RH was measured in case of the most extreme condition ("Max. $CO_2$ fully booked") at a cabin temperature of 25.9 °C. This corresponds to a moisture content of 6.94 g/kg.

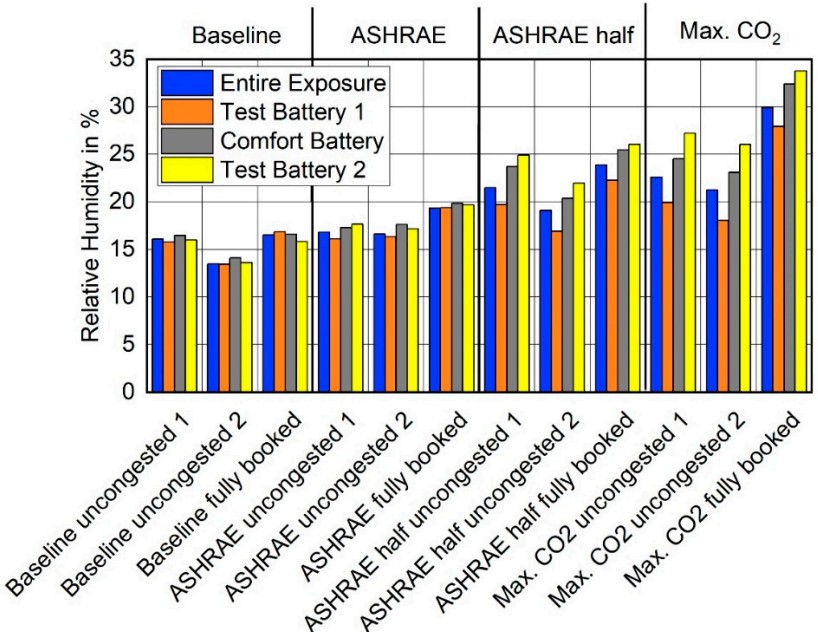

**Figure 9.** Measured average relative humidity in the cabin.

### 3.4. $CO_2$ Concentration

A clear trend of increased cabin $CO_2$ concentration with decreasing outside airflow rate is visible (Figure 11). For the baseline case, the content was 1660 to 1721 ppm (pressure corrected) at the end of the test. For the "ASHRAE half" and "Max. $CO_2$" cases, a gradual increase over the test time is obvious (Figure 12), similar to the findings for relative humidity and with a maximum $CO_2$ level in case of "Max. $CO_2$" between 3853 and 4520 ppm.

### 3.5. TVOCs

TVOC detection was performed by grab sampling and off-line analysis by GC-MS. Results are shown in Figure 13. The general tendency to increased TVOC levels with lower outdoor airflow rate is obvious, however less expressed than for humidity and $CO_2$. The peak in the "ASHRAE half–fully booked" is due to a peak in ethanol. Even though the subjects were told only to drink water and were observed by the cabin crew, it cannot be excluded that somebody brought alcoholic beverages on board. The peak in the "Max. $CO_2$–uncongested 2" condition is due to an event that made it necessary to disinfect the galley with isopropanol cleaning agent during the test conduction. Generally, VOCs are

below or within the limits of UBA [4] level 2 (300–1.000 μg/m$^3$, "no relevant objections, increased ventilation recommended").

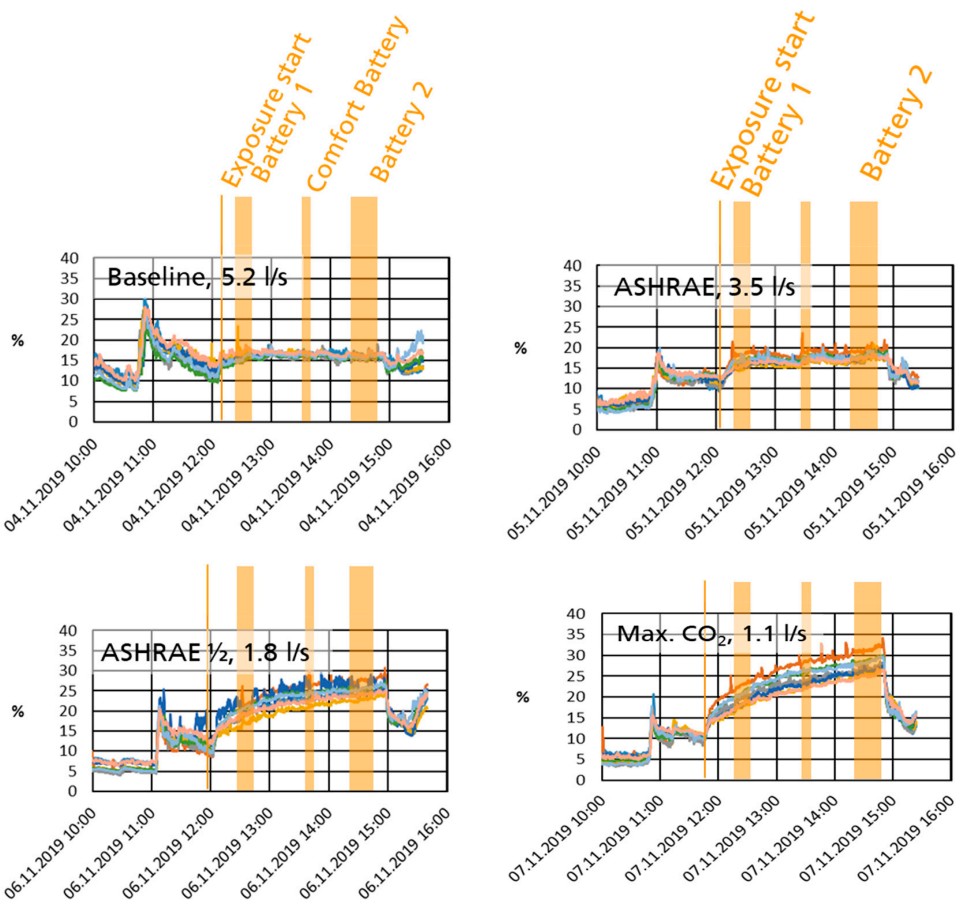

**Figure 10.** Transient evolution of relative humidity in the cabin for four different test days with different outside airflow rates.

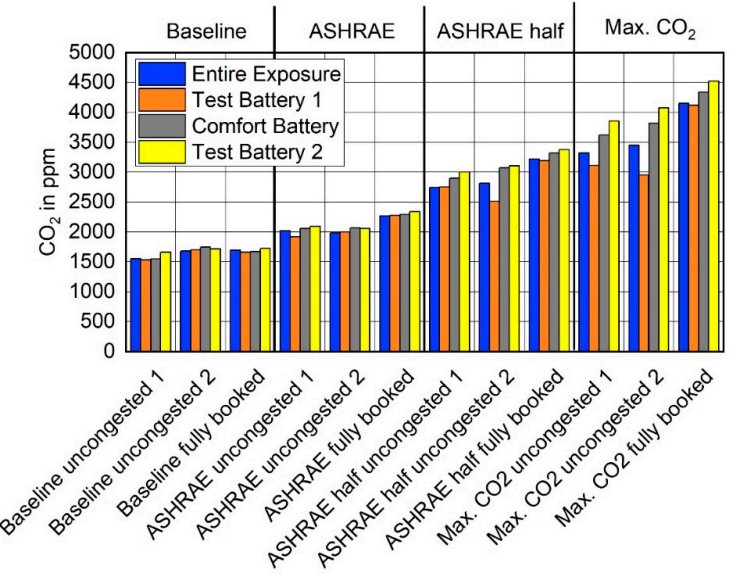

**Figure 11.** Measured average $CO_2$ concentration in the cabin (pressure corrected).

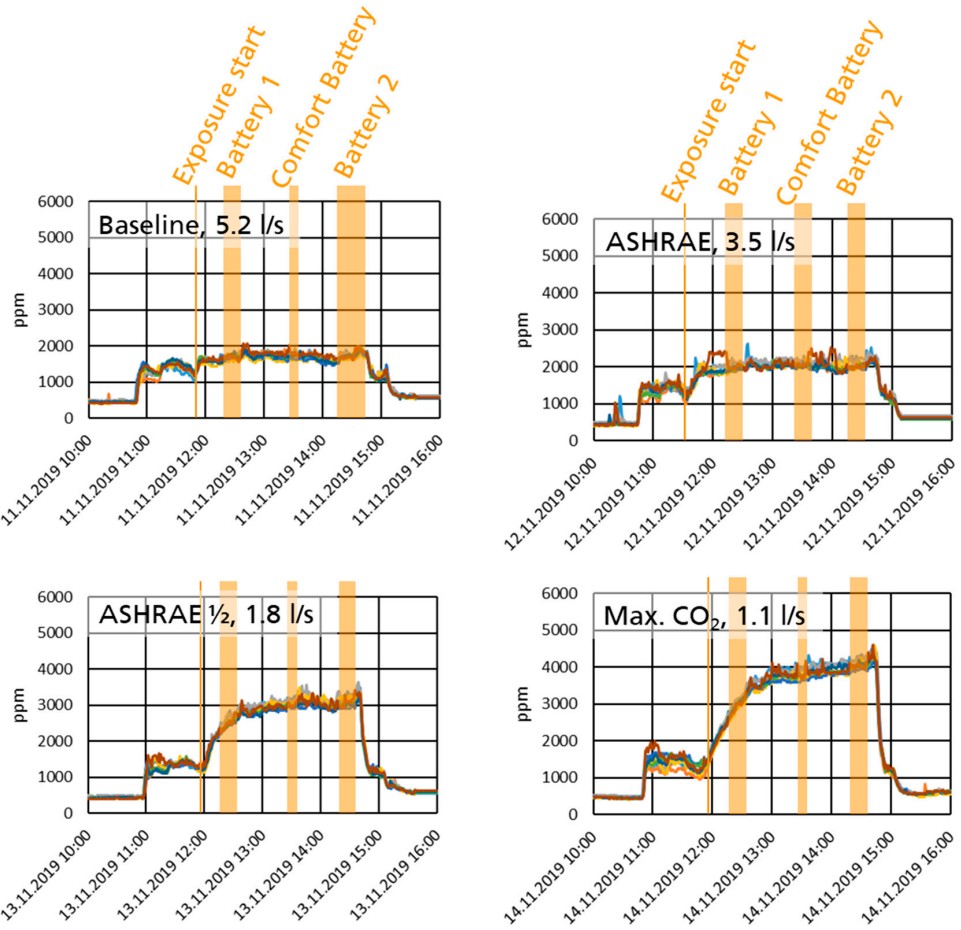

**Figure 12.** Transient evolution of $CO_2$ concentration (pressure corrected) in the cabin for four different test days with different outside airflow rates.

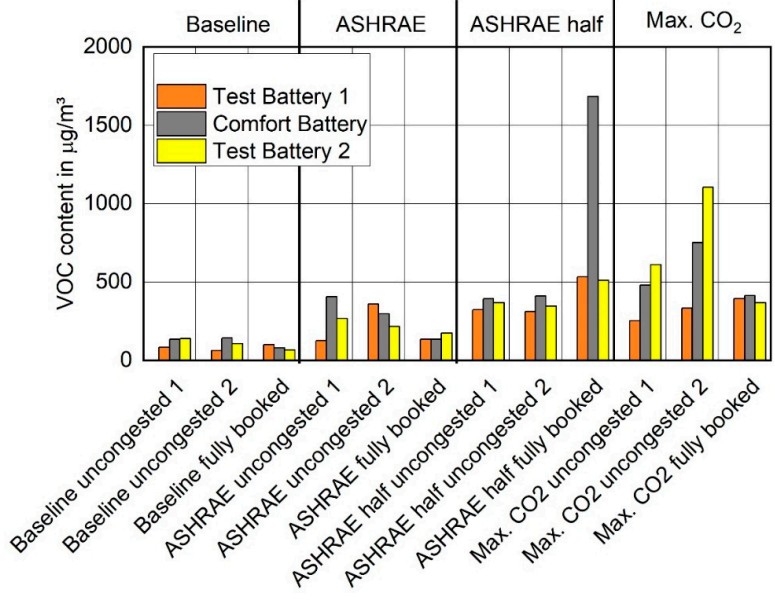

**Figure 13.** Measured TVOC concentration in the cabin air.

In order to assess whether VOCs could have been absorbed by condensed water in the recirculation heat exchanger, the drainage valve was opened after each test. Water was never observed and thus it was concluded that no major condensation occurred.

### 3.6. Trained Panel Votes

The panel of eight trained persons was used in the cases "Baseline uncongested 1", "ASHRAE uncongested 1", "ASHRAE half uncongested 1" and "Max. $CO_2$ uncongested 1". The panel entered the cabin after the comfort battery, slowly walked up and down the aisle and then left to independently give their votes. Through this short residence time, odor adaptation is avoided. The result shows no clear trend to increased perceived intensity or worse hedonic tone. The votes show a distinct intensity of smell and a slightly unpleasant tone (Figure 14).

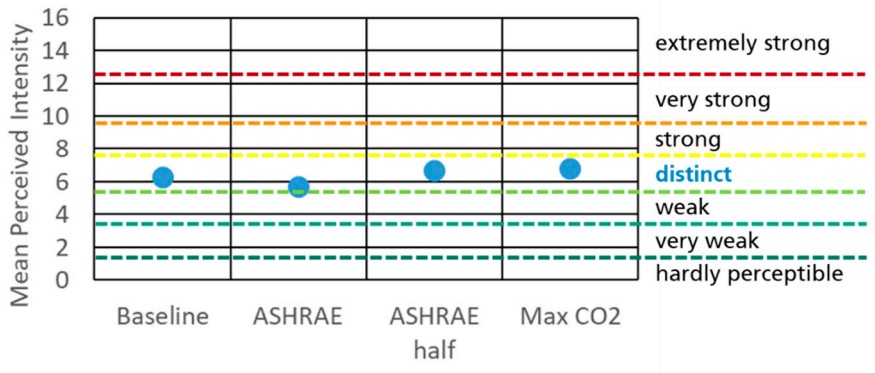

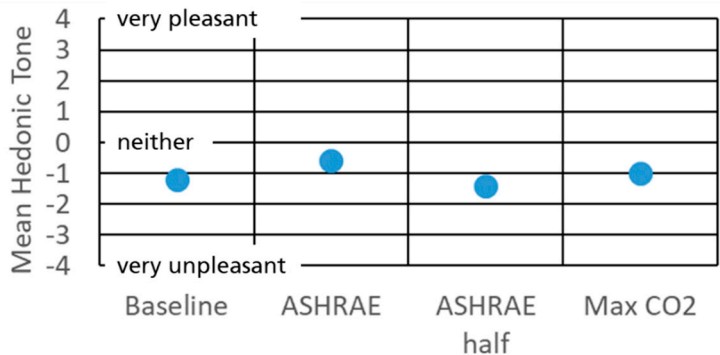

**Figure 14.** Assessment of air quality by trained odor panel.

### 3.7. Subject Votes

In all test cases, subjects were asked at the end of the exposure to assess the smell in the cabin and the acceptability of air quality. For the "Baseline" and "ASHRAE" conditions, the votes on smell are quite similar for the uncongested (Figure 15 left) and the fully booked case (Figure 15 right). For "ASHRAE half" and "Max. $CO_2$", the fully booked condition shows less votes for "no smell" and more votes for "slight" and "moderate" smell perception whereas for the uncongested condition, no systematic difference is found. Analyses of variance (ANOVA) confirm this: there is no main effect of ventilation regime but a small main effect of occupancy (F = 2.99, $p$ = 0.08), interaction of both is not significant (F = 1.72, $p$ = 0.16). That is, the slightly higher level of "slight" and "moderate" smell perceptions in the fully booked conditions with lower outdoor airflow rates are sufficient for an overall difference in perception between uncongested and fully booked sessions, although general levels of smell perception are very low. A similar pattern is found for the acceptability of air quality. However, analyses of variance for this variable show rather clear effects; a decrease of acceptability with decreasing outdoor airflow rate (main effect ventilation regime: F = 3.36, $p$ = 0.019), lower acceptability in the fully booked condition

(main effect occupancy: F = 5.68, *p* = 0.017) and an interaction effect between both (F = 3.97, *p* = 0.008), denoting that only the fully booked condition shows a linear decrease in the acceptability ratings. Figure 16 presents estimated marginal means from this analyses. Nevertheless, the overall acceptability is higher than 75% in all tests performed, and in the dichotomous approach by Wargocki [18,19] more than 93.4% of all participants rated the air quality as acceptable (4.1% inacceptable, 2.5 missing).

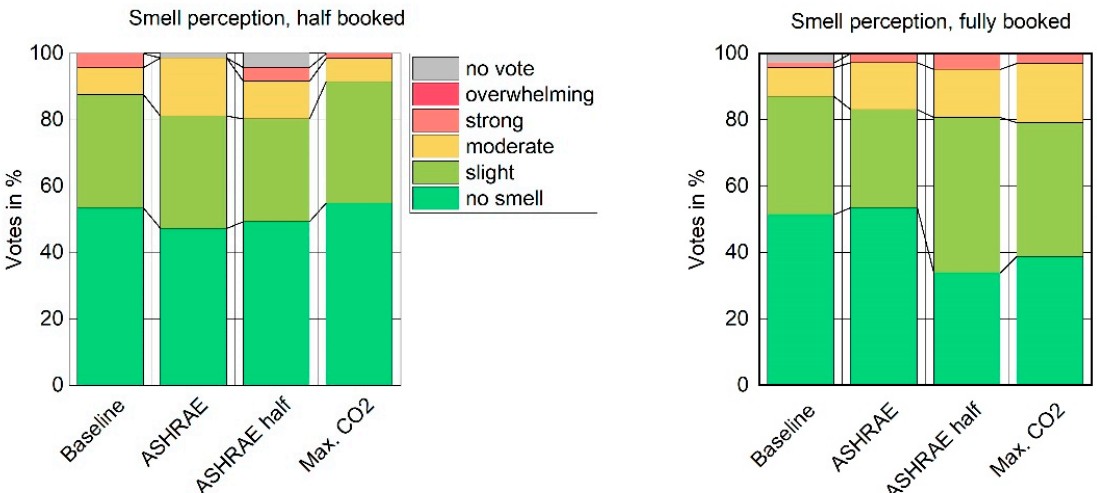

**Figure 15.** Smell perception votes for the different outdoor airflow rates. **Left**: uncongested, **Right**: Fully booked.

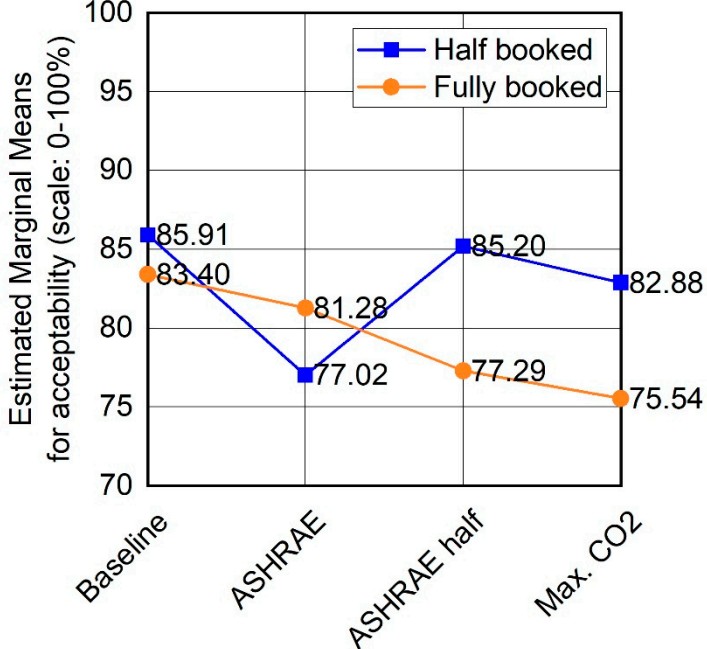

**Figure 16.** Acceptability ratings on a scale from 0–100 controlled for age, sex, smoking and self-reported multiple chemical sensitivity of participants.

## 4. Discussion

This paper shows the effect of lower outdoor airflow rate on cabin relative humidity, $CO_2$ concentration and TVOC level. The measurements were taken during a subject study with a randomized controlled design, blinded participants, representative of flight passengers with regard to age and sex. Thus, the emitted and measured species should be representative for the flying population.

The measured data for the "Baseline" condition show to realistically replicate humidity and $CO_2$ levels reported from commercial flights. The TVOC measurement may be impacted by the used mock-up and the strict behavior rules for the subjects. The votes on perceived intensity, hedonic tone, smell perception and acceptability thus must be critically considered with regard to generalizability to "normal" flights. Moreover, results can only be generalized to spaces where the occupants generate the major emission of VOCs (bioeffluents).

Nevertheless, the principle of an adaptive ECS operation was successfully proven in this study. Here, the airflow rate was adjusted to the number of passengers. Future research should be performed to identify markers in the cabin air that could be used as feedback signal for a controller. Such a marker could be $CO_2$, however it would not cover e.g., the event of increased TVOC found by cleaning requirement or the potential consumption of alcohol as detected in the cabin during this study.

Airflow rates were selected to generate different levels of $CO_2$. Whether these airflow rates are compatible with other requirements of the aircraft ventilation like e.g., exhaust airflow rates in the lavatories and galley, cabin pressurization, cooling of avionics, etc. has been disregarded.

The general trend, to have a higher satisfaction with the environment despite a similar exposure in the uncongested case compared to fully booked one was even reported for other comfort parameters in the "Baseline" case, whereas the other three airflow regimes are currently under evaluation [21].

## 5. Conclusions

This paper presents the indoor air measurements for different outdoor airflow rates in a realistic cabin mock-up with subjects in a simulated flight. The major results are:

- Relative humidity, $CO_2$ and TVOC clearly increase with decreasing outdoor airflow rate
- Singular effects like an ethanol or cleaning agent event showed higher impact on the TVOC levels than the airflow regime
- Neither a trained sensory panel nor subjects could differentiate smell or acceptability for the different airflow conditions. Only in a fully booked cabin slightly worse votes were given at lower outdoor air intake.
- Low outdoor airflow rates necessitate additional cooling capacity in the recirculation path. This would result in a possible need for redesign of the ECS compared to today's architecture.

**Author Contributions:** Conceptualization, V.N. and B.H.; methodology, V.N., F.M., B.H., F.L. and P.W.; validation, V.N., B.H. and P.W.; formal analysis, V.N., F.M., B.H. and R.S.; investigation, V.N., F.M. and R.S.; resources, V.N., F.M., B.H. and F.L.; data curation, V.N., B.H. and P.W.; writing—original draft preparation, V.N.; writing—review and editing, V.N., F.M., B.H. and P.W.; visualization, V.N.; supervision, B.H.; project administration, B.H.; funding acquisition, V.N. and B.H. All authors have read and agreed to the published version of the manuscript.

**Funding:** This study received funding from the Clean Sky 2 Joint Undertaking under the European Union's Horizon 2020 research and innovation programme under grant agreement No. 820872-ComAir-H2020-CS2-CFP07-2017-02.

**Institutional Review Board Statement:** The ComAir study was conducted according to the guidelines of the Declaration of Helsinki, and approved by the Ethics Committee at the Faculty of Medicine, Ludwig-Maximilians-University, Munich (ID: 19-256) on 17 June 2019.

**Informed Consent Statement:** Written informed consent was obtained from all subjects involved in this study.

**Data Availability Statement:** Some data presented in this study are available upon request from the corresponding author. Some data are not publicly available due to contractual restrictions and GDPR regulations regarding privacy for human subject studies.

**Acknowledgments:** We would like to thank Ivana Ivandic and Ines Englmann for their help with the subjects.

**Conflicts of Interest:** The authors are responsible for the content of this publication. The authors declare to have no conflict of interest.

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
