# Peer review of "Effect of Increased Cabin Recirculation Airflow Fraction on Relative Humidity, CO2 and TVOC"

_aerospace, doi:10.3390/aerospace8010015_

Round 1

Reviewer 1 Report

Line 26: ...keeping the cabin at a comfortable temperature

Line 28: Aircraft is a collective noun.  There is no such thing as aircrafts.  "...commercial aircraft the outdoor..."

Line 29: "...upstream of the combustion chamber,..."

Line 32: "...the air supply..."

Lines 41 to 43: Please replace the last sentence on this page.  Suggested replacement: "Aircraft ECS are required to comply with target values for both pressure and temperature.  Typical values for the requirements are listed below."

Line 53 and 54: "...airflow rate to that actually required, thereby avoiding..." 

Line 62: A series of CO2 measurements taken during commercial flights by Giaconia et al., found that average......[8].

Line 125: Figure numbered incorrectly.

Line 127: "Test Matrix and Sequence" - use capital letters in title as for section 2.1.

Line 142: At this point in the paper, it would have been good to have explained why you chose the particular tests that you did.  For example, what evidence did you use to decide to look at ASHRAE half?

From this point onwards, I found the paper very readable.

Author Response

Dear Reviewer,

Thank you very much for your review. We have now adapted the paper with your corrections as suggested for lines 26-62. Additionally, in order to acknowledge the second reviewers comment, we corrected language in the onward text on some further places.

The second figure has been renumbered.

Explanation for ASHRAE half has been given (line 142).

Some of the figures were remade with another software because the second reviewer noticed that resolution sometimes was too low. The data contained is the same, we just made some figures with Origin (allowing to export jpg) rather than Excel.

Attached, please find the Word file with change track. Additionally, we will try to upload a clean Word file in the submission platform.

We wish you a nice Christmas time and a healthy start into 2021!

Best regards,

Victor Norrefeldt & my co-authors

Reviewer 2 Report

Nice work.

Please go over the paper and work on the grammar of some sentences. I corrected some at the beginning of the literature and decided to leave the rest for you. See attached file.

Also, some pictures have low resolution such as Figure 6; the legend for Figure 3 is too small.

References should start with [1] for the first citation in the text. See attached PDF for more details.

Author Response

Dear reviewer,

thank you for your feedback to the paper. We proof read it once again and tried to correct language.

We made some of the figure with low resolution with another software now (Origin and Corel rather than PowerPoint and Excel). Through this, we could provide the major figures in better resolution. The data contained remain the same.

References were renumbered to start with [1] rather than having the alphabetically.

We would like to wish you a merry Christmas time and a healthy start into the new year.

Attached please find the artice with change track. Additionally, we will try to upload a clean version in the submission portal.

Best regards,

Victor Norrefeldt & my co-authors.
